# Large-Scale Proteomic Analysis of Follicular Lymphoma Reveals Extensive Remodeling of Cell Adhesion Pathway and Identifies Hub Proteins Related to the Lymphomagenesis

**DOI:** 10.3390/cancers13040630

**Published:** 2021-02-05

**Authors:** Kamila Duś-Szachniewicz, Grzegorz Rymkiewicz, Anil Kumar Agrawal, Paweł Kołodziej, Jacek R. Wiśniewski

**Affiliations:** 1Department of Pathology, Wrocław Medical University, Marcinkowskiego 1, 50-368 Wrocław, Poland; 2Flow Cytometry Laboratory, Department of Pathology and Laboratory Diagnostics, Maria Skłodowska-Curie National Research Institute of Oncology, Roentgen 5, 02-781 Warsaw, Poland; grzegorz.rymkiewicz@pib-nio.pl; 32nd Department of General and Oncological Surgery, Wrocław Medical University, Borowska 213, 50-556 Wrocław, Poland; dranilpreeti@gmail.com; 4Division of Pathology, Sokołowski Hospital Wałbrzych, Sokołowskiego 4, 58-309 Wałbrzych, Poland; hp2010@pro.wp.pl; 5Biochemical Proteomics Group, Department of Proteomics and Signal Transduction, Max Planck Institute of Biochemistry, 82152 Martinsried, Germany; jwisniew@biochem.mpg.de

**Keywords:** follicular lymphoma (FL), label-free quantitative proteomics, protein-protein interaction network (PPIN), differentially abundant proteins (DAPs), hub proteins, cellular adhesion molecules (CAMs)

## Abstract

**Simple Summary:**

Follicular lymphoma represents the major subtype of indolent B-cell non-Hodgkin lymphomas, ranging from about 20 to 30% of all B-NHLs cases in western countries. Yet, the global proteome profile of follicular lymphoma remains largely undocumented; thus, we aimed to employ for the first time a comprehensive proteomic analysis to outline its molecular landscape. A total of 15 lymphoma fine-needle aspiration biopsy samples and 14 controls were evaluated by label-free quantitative proteomics. Among the 7673 proteins identified in our dataset, 1186 proteins were differentially expressed between lymphoma and control samples. Importantly, dysregulated proteins were enriched in biological processes such as B-cell receptor signaling pathway, cellular adhesion molecules pathway, or membrane trafficking. Additionally, we identified several novel hub proteins related to lymphomagenesis. To summarize, we have determined the molecular characteristics of follicular lymphoma and discovered proteins which may hold potential for biomarkers or therapeutic targets.

**Abstract:**

Follicular lymphoma (FL) represents the major subtype of indolent B-cell non-Hodgkin lymphomas (B-NHLs) and results from the malignant transformation of mature B-cells in lymphoid organs. Although gene expression and genomic studies have identified multiple disease driving gene aberrations, only a few proteomic studies focused on the protein level. The present work aimed to examine the proteomic profiles of follicular lymphoma vs. normal B-cells obtained by fine-needle aspiration biopsy (FNAB) to gain deep insight into the most perturbed pathway of FL. The cells of interest were purified by magnetic-activated cell sorting (MACS). High-throughput proteomic profiling was performed using liquid chromatography-tandem mass spectrometry (LC-MS/MS) and allowed to identify of 6724 proteins in at least 75% of each group of samples. The ‘Total Protein Approach’ (TPA) was applied to the absolute quantification of proteins in this study. We identified 1186 differentially abundant proteins (DAPs) between FL and control samples, causing an extensive remodeling of several molecular pathways, including the B-cell receptor signaling pathway, cellular adhesion molecules, and PPAR pathway. Additionally, the construction of protein–protein interactions networks (PPINs) and identification of hub proteins allowed us to indicate the key player proteins for FL pathology. Finally, ICAM1, CD9, and CD79B protein expression was validated in an independent cohort by flow cytometry (FCM), and the results were consistent with the mass spectrometry (MS) data.

## 1. Introduction

Mature B-cell non-Hodgkin lymphomas (B-NHLs) represent the majority of diagnosed lymphoma and are a very diverse set of malignancies, including differences in the cell of origin, aggressiveness, and response to therapy [1,2,3]. The most frequent of indolent B-NHL subtypes is follicular lymphoma (FL) [4], ranging from about 20 to 30% of all B-NHLs cases in western countries [5]. However, our recent study confirmed a lower FL incidence rate in Poland as compared to other European countries. FL accounted for 6.3% of all B-NHLs, with 5.6% in the male and 6.9% in the female populations, respectively [6].

FL is generally considered a slow-growing lymphoma, often relapsing with a low malignancy degree, and slow progression is common during the natural history of the disease [7,8]. Some cases present with the rapidly histological transformation (HT) to the more aggressive lymphoma, mostly diffuse large B-cell lymphoma, not otherwise specified (DLBCL, NOS). Less frequently FL transforms to the high-grade B-cell lymphoma (HGBL) with MYC and BCL2 and/or BCL6 gene rearrangements, also known as “double-hit” or “triple-hit” lymphomas [9,10]. The HT rate of FL is difficult to estimate [11] and it is assumed about 2–3% per year from diagnosis [12,13,14]. Recently, significantly prolonged progression-free survival was observed when treated with modern therapies [15,16], however its remains incurable for the majority of elderly individuals [6,17,18]. Therefore, extensive “omics” studies dealing with the molecular aspect of FL are highly recommended. Among the techniques for studying proteins, proteomics is the most advanced aiming at a simultaneous investigation of thousands of differentially abundant proteins (DAPs) in one sample and, together with bioinformatical tools, reveal alterations in the biological pathways.

Attempts to characterize the B-NHLs by proteomic profile have been previously proposed but have not yet been translated into clinical practice. Few mass spectrometry (MS)-based proteomic studies with different designs regarding samples and analytical methods were previously performed, mainly on diffuse large B-cell lymphoma (DLBCL) [19,20,21,22]. However, most of the above research relies on the use of lymphoma cell lines, and studies engaging a representative set of a clinical sample are still rare. Surprisingly, there is a lack of deep global proteomic data regarding FL resulting from an analysis of clinical samples, and the current molecular landscape of disease bases mostly on genomic studies [23,24]. The available data comprises only a few mass-spectrometry based research performed several years ago, mainly on cell lines [25,26,27]. Additionally, a majority of previously published works have been performed using two-dimensional gel electrophoresis protein separation and further MS analysis of dissected spots which is a semi-quantitative method; hence, it does not deliver information about the accurate protein concentration.

First attempts at studying the FL proteomic profile were made by Weinkauf et al. in 2007 [26]. Authors compared three FL and three mantle cell lymphoma (MCL)-derived cell lines and identified 1600 protein spots per gel, of which 175 were found to be differentially abundant. Additionally, authors made a comparison of proteome analysis and RNA expression array data. In 2010 Jansen et al. [27] performed MS-based protein profiling on a large group of 239 B-NHL samples, including 63 FL cases. The lymph node biopsies were obtained from multiple hospitals in accordance with standard pathology protocols, not pre-isolated before MS. The authors successfully classified B-NHL subtypes using two independent biostatistical methods; however, this study did not focus on molecular biology and pathway analysis. More recently, Rolland et al. [28] evaluated global phosphoproteomic signatures of B-NHLs across 11 cell lines of three distinct lymphoma subtypes, including FL, MCL, and Burkitt lymphoma. Authors identified 1701 unique phosphorylated proteins and observed that phosphoproteomic signatures are associated with each lymphoma subtype. These data provide a detailed annotation of phosphorylated proteins in human lymphomas. In turn, the study of Ludvigsen et al. 2015 [29] has shed some light on the histological transformation within FL, revealing the differentially expressed protein profiles of FL obtained from patients with diverse clinical outcomes.

Intrigued by the observation that the global proteomic profile of FL remains largely undocumented, we first employed a comprehensive proteomic analysis of patients derived samples vs. normal B-cells. We analyzed the global proteome of 15 follicular lymphomas and 14 control samples using LC-MS/MS. Notably, cells of interest were obtained by FNAB, their quantity and quality have been confirmed by flow cytometry (FCM) and cytogenetics, and pre-sorted by magnetic beads to analyze pure B-cell population.

Using the ‘Total Protein Approach’ (TPA) [30,31], we generated precise information about the accurate concentration of proteins in the studied samples. LC-MS/MS analysis resulted in the identification of 6724 proteins detected in at least 75% of study groups, of these 579 proteins have increased abundances with lymphoma/control fold change ≥2.0, while 607 proteins were down-represented with lymphoma/control fold change at least 2-fold lower. Next, we conducted a comprehensive enrichment analysis with Perseus and Cytoscape software and we identified several enriched pathways, including cellular adhesion molecules’ (CAMs) interactions, which possibly perform important roles in FL. A key signaling pathway involved in the development of B-NHL lymphomas, the B-cell receptor (BCR) signaling pathway, was also identified as significantly changed. Notably, numbers of hub proteins, essential for new drug development for pathway-based targeted therapy, were identified. Our results describe, to the best of our knowledge, the largest functional analysis of the in-depth proteome of follicular lymphoma.

## 2. Results

### 2.1. General Description of Analyzed Material

Considering the identification of only proteins with at least two peptides, we found a minimum of 6832 and a maximum of 7383 proteins per sample. The concentration of identified proteins ranges from about 50 fmol/mg (e.g., MAP3K1, C9orf72, SLC23A2) to about 2.7 nmol/mg (e.g., H3F3A). A total of 7673 protein groups were identified and quantified. Appendix A provides details of all proteins identified in this study, including concentration values (pmol/mg) in each sample. 6724 proteins were detected in at least 75% of study groups, of these 579 proteins were overrepresented with lymphoma/control fold change ≥2.0, while 607 proteins were downrepresented with lymphoma/control fold change at least 2-fold lower, both in combination with a *p*-value ≤0.05. 201 proteins were present at more than 4-fold higher concentrations in lymphoma samples, among which 58 proteins with fold change ≥7. In turn, 267 proteins had at least 4-fold lower concentration and 47 proteins exhibited fold change ≤0.1. Appendix A include details of all 1168 DAPs; FL/C ratios, and Student’s *t*-test statistics.

Hierarchical clustering of the top 454 DAPs identified by Student’s *t*-test was performed in Perseus software and revealed distinct protein profiles between lymphoma and control samples, Figure 1A. Next, Perseus was also used to generate a volcano plot, with a fold change cutoff of 2, Figure 1B. Protein grouped in the right and left corner were considered significantly up- and down-regulated, respectively.

### 2.2. Identification of DAPs 

The proteins that show a differential abundance in each group may potentially underscore the phenotypic shift in B-cells towards FL. The top 20 up- and downrepresented proteins with the highest differences in concentrations among the study group are presented in Table 1. 

Of the 579 overrepresented proteins, six proteins were found in at least 20-fold higher abundance in lymphoma patients (NUCKS1, SLC14A1, GPALPP1, ADPRH, CD72, PRDM15). Of these proteins, the NUCKS1 showed the biggest expression difference between lymphoma and normal B-cells (36.53-fold upregulation). In turn, three proteins (ITGAV, NT5E, and PLTP) out of 607 downrepresented proteins were found in at least 20-fold lower concentrations in FL. Notably, Search Tool for the Retrieval of Interacting Genes/Proteins (STRING) analysis performed on the proteins with the highest abundance decreases with lymphoma/normal ratio ≤0.01 revealed that 16 out of 32 identified proteins interacted with each other in a protein-protein interaction network, Figure 2A.

To illustrate the interaction of all 1186 dysregulated proteins, a PPIN was constructed using the STRING database. 1139 nodes and 7543 interactions between them were found.. Next, the hub proteins were selected from the PPIN. Here, we used the cytoHubba plugin based on Cytoscape software for ranking proteins in a network by their network features [32]. In line with the previous study [33], three topological methods were used to find out the hub proteins, including maximal clique centrality (MCC), maximum neighborhood component (MNC), and Degree algorithm. Venn diagrams were used to show the shared sets of proteins. The overlapping proteins were considered to be more reliable. The results showed that four of the 20 hub proteins were shared between three methods, including three up-regulated proteins (TGOLN2, CBL, SH3KBP1) and one down-regulated protein; RPS27A, Figure 2B. In turn, Figure 2C shows the PPIN of the top 20 hub proteins selected from all DAPs according to the Degree algorithm. Notable, 6 of the 20 hub proteins were engaged in adhesion processes: FN1, CTNNB1, CDH1, ITGB1, ICAM1, and ITGAV. Among these proteins, ITGAV exhibited the most significant difference between the lymphoma and control samples.

### 2.3. KEEG Pathway Analysis

Through the use of Perseus, all identified confidential proteins were annotated for Kyoto Encyclopedia of Genes and Genomes (KEGG) and the number of DAPs involved in selected pathways was calculated, Figure 3. Surprisingly, the most significantly changed pathways were those associated with adhesion, including ECM-receptor interaction, cell adhesion molecules, GAP junctions, focal adhesion, and adherence junctions. Furthermore, the results revealed that the DAPs in FL were associated with several key pathways constituting no less than 25% of the identified proteins, including B-cell receptor signaling pathway, leukocyte transendothelial migration, phosphatidylinositol signaling system, Notch signaling pathway, and PPAR signaling pathway. Signaling through these pathways during the development of FL has been extensively discussed in the literature [34,35]. Moreover, the percentage of proteins involved in apoptosis and glycolysis/gluconeogenesis was also relevant. In turn, the percentage of DAPs in the metabolic pathway related to oxidative phosphorylation, DNA replication, basal transcription factors, and cell cycle was below 10%. These observations could explain the relatively low proliferation rate endurance of indolent FL cells in comparison to more aggressive B-NHLs.

### 2.4. Protein-Protein Interaction Networks and Hub Proteins in FL

Four pathways (BCR signaling, CAMs pathway, PPAR signaling pathway, and pathway in cancer), closely related to B-cell functions, were investigated in more detail using the online STRING database. Next, a protein interaction relationship network table was downloaded and visualized using the STRING plugin of Cytoscape software, Figure 4A–C.

#### 2.4.1. The DAPs in BCR Pathway

BCR is the surface receptor for antigens in mature B-cell and BCR signaling is essential for normal B-cell development and maturation [36]. In this study, 57 proteins related to the B-cell receptor pathway were identified by LC-MS/MS, of these 55 were found to be functionally connected according to the STRING database, Figure 4A. 580 functional links between proteins were found. Regarding the *t*-test data from Perseus, 18 (31.5%) of the proteins were found to be up-regulated (red), while only PIK3CA was found to be down-regulated (green). Furthermore, RASGRP3, BLNK, and CD79B were found to be close to the center of the network with high fold changes of 11.78, 5.22, and 4.25, respectively. Next, we used the MCC, MNC, and Degree algorithms of cytoHubba to identify the top 10 hub proteins from all proteins enriched in the B-cell receptor pathway, Appendix A. Overlapping hub proteins, which were found to be significantly changed in FL are VAV1, LYN, BTK, and PIK3CA.

#### 2.4.2. The DAPs in CAMs Pathway

The network made of 51 proteins involved in cellular adhesion with 403 links was found within our dataset. Figure 4B shows an interaction network between 51 cellular CAMs identified in this experiment. Five of them had increased abundance in FL (ICAM1, ICAM3, NCAM1, CD22, and CD86), whereas 12 were underrepresented (CDH1, PTPRF, ITGAV, ITGA6, ITGB1, CNTN1, F11R, MPZL1, SIGLEC1, CLDN3, SDC4, and CD8A). Importantly, several of DAPs identified in this study were well-known key players in the cell-cell and cell-ECM interactions, including ITGB1, CDH1, CNTN1, and ICAM1. Most of the DAPs are located peripherally and usually create less than 10 interactions. The exception is ICAM1, which is functionally connected with 27 other CAMs, and CD86, which interacts with 25 proteins. CDH1, ITGAV, and ICAM3 are functionally connected with the other 18 proteins. The cytoHubba analysis of hub proteins indicated that ICAM 1 and CD86 proteins performed a central role in the CAMs protein network, Appendix A.

#### 2.4.3. The DAPs in PPAR Signaling Pathway

Peroxisome proliferator-activated receptors (PPARs) are a family of nuclear receptors that regulate lipid metabolism [31,37]. In our investigations, the PPAR pathway is represented by 30 proteins. Of these, 24 were found to be functionally related with 140 links, Figure 4C. Interestingly, eight network belonging proteins (30%) were found to be significantly down-regulated, including PLTP and PCK2 with the highest underrepresentation of 0.05 and 0.18, respectively. Regarding the analysis performed with cytoHubba, three down-regulated proteins (ACOX1, ACSL3, and SLC27A1) were found to be key players in the PPAR pathway, Appendix A.

#### 2.4.4. Pathway in Cancer Analysis

Pathway in cancer (hsa05200) was found to be the top identified pathway dysregulated in FL amongst our dataset. KEGG pathway results showed that 142 proteins were mainly enriched in the pathway in cancer. Thus, this pathway was the most enriched in this study, and we decided to analyze it in more detail. We uploaded all identified proteins to STRING to generate the PPIN. 138 of 142 proteins were found to be functionally linked by 2393 connections, Figure 5A.

In this group, we found 30 significantly overrepresented proteins and 25 underrepresented proteins that together yield a high percentage of 55/142 (38%) of differentially expressed proteins. Importantly, the group of up-regulated proteins comprises the well-known key player proteins in cancer biology, e.g., WNT (fold change = 10.79), TRAF5 (fold change = 6.75), MAP2K1 (fold change = 5.51), (BRAF, fold change = 4.25), and caspase 9, caspase 3 and caspase 8 with fold changes of 4.69, 4.22, and 3.31, respectively. In Figure 5A, we marked the proteins belonging to the mTOR signaling pathway, MAPK signaling pathway, and chemokine signaling pathway. Next, in accordance with the Degree algorithm of cytoHubba, we identified the hub proteins in pathway in cancer within all identified proteins (Appendix A), and among the DAPs (Appendix A). Figure 5B,C illustrates the functional network of identified hub proteins in pathway in cancer.

### 2.5. Pathway Enrichment Analysis by ClueGO

#### 2.5.1. Up-Regulated Proteins in FL

The up-regulated and down-regulated proteins in FL were uploaded to the ClueGO application of Cytoscape and analyzed separately. KEGG, Reactome, and WikiPathways databases were selected with statistical criteria set at *p* ≤ 0.05, and the pathway terms were ranked based on the fold enrichment. Pathway enrichment of 579 significantly up-regulated proteins revealed that they were classified into 28 pathways within the seven function cluster groups and involved mainly in the B-cell receptor signaling pathway, signaling by interleukins, membrane trafficking, and immune system, Figure 6A and Appendix A.

The most abundant group of pathways functionally enriched in this study includes those related to BCR signaling. Data visualization showed that 16 up-regulated proteins were assigned to the BCR signaling in accordance with KEEG (KEGG:04662), while 24 proteins following Reactome (R-HSA:983705) and WikiPathways (WP:23). In turn, the most abundant pathways functionally enriched in this study included those related to membrane trafficking (R-HSA:199991) and counted 46 proteins, Appendix A. Besides, 22, 7, and 14, proteins overrepresented in FL were associated with EGFR signaling pathways (WP:437), signaling by erythropoietin (R-HSA:9006335) and dectin-1 signaling (R-HSA:5607764), respectively. The complete list of up-regulated proteins, which contributed to the all enriched pathway, are listed in Appendix A.

#### 2.5.2. Down-Regulated Proteins in FL

ClueGO Cytoscape application assigned the 607 down-regulated proteins in this study to the 21 functional annotation groups with 61 pathways, of which six functional groups were represented by three or more pathways, and 15 main groups were represented by one or two pathways, Figure 6B and Appendix A. The enrichment of proteins with decreased abundances in our dataset mainly affected non-integrin membrane-ECM interactions, regulation of complement cascade, cell-cell communication, protein localization, interleukin signaling pathway, and extracellular matrix organization. The downrepresented proteins, which contributed to the all enriched pathways are listed in Appendix A. The most abundant functional group contains 12 terms related to non-integrin membrane ECM-interactions, Appendix A.

### 2.6. Flow Cytometric Validation of CD9, ICAM1 and CD79B Expression

FCM was used to analyze the expression of the selected proteins: CD9, ICAM1 (CD54), and CD79B, which were differentially abundant in our proteomic study. One representative example of FNAB/FCM analysis is presented in the Appendix A. In general, the expression level of CD9, CD79B, and ICAM1 validated by FCM was in line with the results obtained by LC-MS/MS, Appendix A.

## 3. Discussion

Proteins interact with each other creating functional pathways. In general, the degree of connectivity of a certain protein influences the entire cell proteome and may lead to system failure when the protein changes. Protein-protein interactions and the resulting networks perform a pivotal role in the biological processes, including the development of B-NHLs. There is increasing evidence that deregulation of signaling pathways underlies the pathogenesis of hematological and lymphoid malignancies [38]. However, the extent to which differentially expressed proteins may be involved in FL pathogenesis is still fragmentary. Investigation of lymphoma-related proteins in the human PPIN network performed in this work would, therefore, provide valuable information for elucidating the molecular mechanisms of lymphomagenesis. In addition, the important benefit of networking is the ability to select essential proteins in PPIN, the so-called hubs or hub proteins which are defined as the most highly connected central proteins.

During the last decade, there was increasing evidence that BCR signaling has emerged as a central oncogenic pathway that directly regulates growth, survival, and proliferative signals in normal and malignant B-cells [39,40,41] what make this pathway a valuable target for the treatment of B-NHLs. The abovementioned studies indicated that chronic BCR signaling in B-NHLs might partially result from mutations in genes encoding BTK and CARD11. Mutations in BTK and CARD11 were reported in 30% of FL patients [8]. Importantly, both proteins were overrepresented in our dataset. BTK is distinctively expressed in B-cells and is involved in the differentiation and activation of B-cells [36]. It is crucial to the function of BCR signaling pathway, where signals from SYK are received and transduced to initiate downstream signaling pathway [42]. Furthermore, BTK is involved in activating Fc-receptors, thus the upregulation of BTK results in the upregulation of Fc epsilon receptor (FCER) signaling [43], what was also reflected in our work by ClueGO analysis. The exact mechanism of this process needs to be investigated; however, our results show the central role of BTK and SYK in BRC signaling pathway according to an analysis performed with cytoHubba.

On the other hand, B-NHLs frequently harbor genetic mutations leading to abnormal activation of canonical nuclear factor-κB (NF-κB), which is a hallmark of different lymphoma subtypes. It is widely accepted that BCRs activate NF-κB by harboring receptor mutations in CD79A or CD79B, along with mutations of kinase LYN or effector CARD11 [44,45]. CARD11 is the most widely studied signal integrator that translates BCR and TCR triggering the activation of the canonical nuclear factor- κB (NF-κB), and mTOR pathways [46,47]. Importantly, Phelan et al. [48] reported the CARD11, CD79A, and CD79B genes overexpression in 574 DLBCL biopsies. In turn, Myklebust et al. [49] confirmed the elevated CD79B expression in FL cells and its strong correlation with BCR-induced signaling. Of notice, all the protein products of the above genes were found to be overrepresented in our dataset, and the central role of LYN in BCR signaling was established. The expression of CD79B protein was also verified using FCM and results were consistent with the MS data.

Another hub protein identified in this study is VAV1, with a crucial role in receptor tyrosine kinase (RTK) pathway [50]. Interestingly, the VAV1 expression is limited to the hematopoietic system during normal development [51] and at the same time, VAV1 remains overexpressed in hematological and lymphoid malignancies, including chronic lymphocytic leukemia (CLL) [52], acute myeloid leukemia (AML), and DLBCL [53], what is in line with our proteomic results. An expression profiling study on DLBCL cell lines [53] also demonstrated an association between increased VAV1 expression and a higher proliferative activity of lymphoma cells. Importantly, several other proteins belonging to the BCR pathway with a fold change ≥5.0 were also identified, including MAP4K1, BLNK, RAPGEF1, VAV2, MAP2K1, MAP3K7, DAPP1, and SH3KBP1.

PIK3CA is the only protein classified as a hub protein and down-regulated at the same time, while other phosphoinositide 3-kinases (PI3Ks) were frequently overexpressed. PIK3CA encodes the p110α catalytic subunit of PI3K and regulates signaling pathways associated with cell adhesion, proliferation, and survival [54]. The PI3K pathway is considered to perform an important role in lymphomagenesis. The presence of several PIK3CA oncogenic mutations has been reported in DLBCL patients [55], as well as PIK3CA has been investigated for its clinicopathological significance; however, no data have been reported in indolent B-NHLs. Thus, the observed here downregulation of PIK3CA in relevance to its biological role has to be confirmed and then explored on a larger set of clinical samples. Taken together, these proteomic data are in line with widely accepted evidences that the elevated expression of proteins belonging to the B-cell receptor pathway is closely associated with the follicular lymphomagenesis. Moreover, the independent enrichment analysis performed in ClueGO additionally confirmed the extensive remodeling of BCR signaling pathway in FL. Concurrently, we found a number of new hub proteins which were not previously implicated in follicular lymphomagenesis, and visualized for the first time the complex interaction network between them.

The role of cellular adhesion molecules in lymphomagenesis was partially revealed 3 decades ago, with the observations that B-NHL growth and clinical aggressiveness may be related to the adhesive capacities of the tumor cells [56]. Meanwhile, our knowledge about the cancer cell adhesion and ECM-receptor interactions has dramatically increased, however still little is known about CAMs expression in B-NHLs and many reports remain unclear. The value of CAMs expression in FL is fragmentary, and the biological significance of their expression in hematological malignancies controversial, which may partially arise from a poor reproducibility of immunohistochemical staining. The major finding of our study is strong evidence that the CAMs pathway contributes to the follicular lymphomagenesis.

ICAM1, which is a hub protein for lymphoma trafficking, has been shown to be overrepresented in several carcinomas, including breast carcinoma [57,58]. ICAM1, also known as CD54, is established to mediate the adhesion of malignant epithelial cells to the lymphatic endothelium and, therefore, promote the tumor spread in regional lymph nodes [59]. Over 20 years ago, Terol et al. observed that lower expression of ICAM1 in aggressive B-NHL correlates with a more advanced stage of the disease, higher bone marrow infiltration, and worse prognosis [60], however, Schniederjan et al. demonstrated a decreased ICAM1 expression in aggressive BL compared to DLBCL [61].

In line with our reports, the high-throughput sequencing and bioinformatics analyses of several tumors have revealed that CAMs are frequently identified as hub genes based on a PPIN analysis [62,63,64]. These recent studies underline the particular importance of CAMs in tumor development and progression. Lately, Peng with colleagues suggested that cell adhesion might play a crucial part in the malignant progression of multiple myeloma (MM) [62]. They identified 12 hub genes, including ITGB1, FN1, ITGA5, and CDH1, which protein product were identified as hub proteins in our dataset. Additionally, the authors demonstrated that numbers of CAMs are valuable biomarkers for the diagnosis of MM. CD86 is a ligand for the costimulatory receptor CD28 and the coinhibitory receptor CTLA-4 that activate T-cells against antigens presented by antigen-presenting cells [65]. It was suggested that downregulation of CD86 helps cancer cells to escape from the immune attack [66]; however, its higher expression in malignant cells compared to corresponding control cells was observed in some malignancies, such as pancreatic carcinoma [67] and FL. The upregulation of CD86 observed in FL was associated with increased APC activity and enhanced triggered T-cell responses [68]. In turn, Wang et al. confirmed that elevated CD86 expression was associated with a better DLBCL prognosis [69].

Interestingly, in the entire dataset, the ITGAV was the most down-regulated protein. ITGAV is a member of the integrin alpha chain family, which could connect with integrin beta subunits to form αvβ1, αvβ3, αvβ5, αvβ6, and αvβ8 integrin receptors [70]. ITGAV is implicated in tumor migration and invasion; thus, its overexpression has been reported in several epithelial tumors including gastrointestinal tract, prostate, and breast cancer [71,72,73,74]. For breast cancer with high ITGAV expression, it has been recently suggested as a potential drug target [74]. More recent study reported ITGAV upregulation in extranodal DLBCL cells compared to nodal DLBCL, suggesting its role in dissemination and other routes of spread of both DLBCL subtypes [75]. The underlying cause for ITAGAV downregulation confirmed in our dataset is not known. According to Human Protein Atlas [76], 3 out of 6 normal lymph nodes exhibit low ITGAV expression, while 11 out of 12 lymphoma samples were negative. The role of ITGAV definitely needs to be further elucidated and a future study should be carried out to evaluate the expression level of ITGAV and its clinicopathological and prognostic significance in FL patients. However, we speculate that the downregulation of alpha V integrin on the surface of FL cells may reflect the less tumorigenic phenotype of this lymphoma. The other downrepresented cell surface protein is CD9 antigen belonging to the family of tetraspanin. Notably, data validation performed with FCM exhibited overall weaker CD9 expression in FL cells compared to normal B lymphocytes. The previous study had reported the loss of CD9 protein expression in B-NHLs and suggested that CD9 inactivation may perform an important role in B-NHLs transformation [77]. In line with our results, the study of Dong et al. confirmed the downregulation of CD9 compared with normal controls [78]. Moreover, authors suggested that CD9 expression negatively correlated with FL prognostic outcomes and attributed to FL progression.

Above all, these results indicated that the CAMs and ECM-receptor interaction pathway, which were hitherto overshadowed by well-known canonical pathways involved in lymphomagenesis, are taking on greater importance in FL. Indeed, the results obtained in our study indicate the dual activity of CAMs and downregulation of certain cell adhesion molecules can be explained as a lymphoma cell strategy to prevent their recognition by the immune system, as was recently suggested [66]. To the best of our knowledge, a comprehensive analysis of CAMs expression and their networking in B-NHLs has not been undertaken, particularly in FL. In our dataset, the numerous CAMs, rather than a single molecule, were found to be both down- and up-regulated, and the combined function of these molecules may affect the general adhesive properties, what is in accordance with the recent research. However, the exact roles of selected hub CAMs in FL remains to be established. 

Previous studies have demonstrated close interactions between FL cells and the surrounding microenvironment [79]. Notably, FL primary cells were observed to have impaired adhesive properties to both: stromal cells and Matrigel than normal B-cells in our previously published optical tweezers study [80]. Enrichment analysis of down-regulated proteins performed with ClueGO strongly supports their implication in ECM-receptor interaction related processes.

Dysregulated PPAR signaling in many carcinomas is associated with disturbances in multiple metabolic processes [81,82]. Increasing evidence has shown the tumor-suppressive effect of PPAR agonists in several cancer, suggesting that the activation of PPARs could serve as a potential strategy for therapeutic treatment of, e.g., bladder cancer [83]. However, the biological roles and the underlying mechanisms regulated by PPAR signaling pathway in normal and neoplastic B-cell development are not well understood. Recently, an extensive analysis of PPAR pathway genes using integrated genomic, transcriptomic, and clinical data from 18.484 patients involving 21 cancer types was performed [84]. The authors confirmed the downregulation of several genes in at least 7 cancer types, including ACOX3, PKC2, SLC27A4, and SLC27A2. Consistent with this notion, the protein products of all the above genes were found to be significantly down-regulated in our work. Regarding B-cell lymphoma, the PPAR gamma pathway was found to be down-regulated in primary diffuse large B-cell lymphoma of the central nervous system (CNS DLBCL) [85].

In this work we identified several proteins processing in classical developmental signaling pathways as being differentially regulated, which were not discussed in this manuscript. Apart from well-known markers in B-NHLs, many of these proteins may function as key molecules in follicular lymphomagenesis. Thus, we strongly encourage the reader to analyze the data included in Appendix A for further information on the abundance changes and function of the proteins of interest.

## 4. Materials and Methods

### 4.1. Collection of FNAB Samples and Clinical Information

All FL cases were diagnosed and graded (grade 1, 2, and 3A) by a hematopathologist using histopathological and immunohistochemical examinations according to the current 2017 WHO classification [15]. Pre-therapeutic lymphoma samples as cellular suspensions were also obtained by the same hematopathologist by FNAB/ultrasound-guided FNAB from the involved lymph nodes or tumors of patients diagnosed at Maria Sklodowska-Curie National Research Institute of Oncology between 2016 and 2017. Each case, obtained by FNAB, was also followed by detailed FCM immunophenotyping, assessment of % of pathological cells, and cytogenetics analysis as previously described [86]. Subjects submitted for lymphadenopathy screening and with reactive histopathological findings served as controls. Reactive lymph nodes are suitable comparison material for studies of tumor-involved lymph nodes and follicular lymphoma, as previously described [87]. In total, 15 samples from FL and 14 samples obtained from controls were included in high-throughput proteomic analysis. Appendix A shows the clinical characteristics of FL patients and controls. Fine needle aspiration biopsies were collected to a collection tube filled with 10 mL of cold phosphate-buffered saline (PBS) and immediately transferred to K_2_EDTA tubes (EDTA 7.2 mg, Becton Dickinson, Heidelberg, Germany) for 20 min. Next, the FNAB biopsies were snap-frozen in a cryotube in the presence of 30% fetal bovine serum (FBS, Thermo Fisher Scientific, Waltham, MA, USA) and 10% dimethyl sulfoxide (DMSO, Sigma-Aldrich, Steinheim am Albuch, Germany) for further studies. Samples were transported and stored in liquid nitrogen.

### 4.2. Magnetic Isolation

Prior to the sample preparation for mass spectrometry, biopsy specimens were thawed, centrifuged at 2000 RCF for 10 min, and resuspended in RPMI-1640 medium (Thermo Fisher Scientific) with 1% Primocin (InvivoGen, Toulouse, France). Cells were kept in suspension overnight at 37 °C in 5% CO_2_ and then purified with Ficoll-Paque (Sigma-Aldrich). Cells were washed in PBS by centrifugation. Cell debris was removed from the sample by the use of Dead Cell Removal Kit (Miltenyi Biotech, Bergisch Gladbach, Germany) according to the manufacturer’s recommendations. Between each step, cells were washed with cold PBS. Next, B-cells were isolated from mononuclear cells by positive selection using the Miltenyi CD19 Positive Isolation Kit (Miltenyi Biotech), as previously described [88]. Ultimately, the sample purity was consistently over 95% of CD19 positive B-cells exhibiting malignant phenotype and over 98% for normal B-cells what was confirmed by FCM for selected samples. After purification, cells were washed three times with cold PBS, centrifuged at 1200 rpm for 5 min at 4 °C and resuspended in lysis buffer (100 mM Tris hydrochloride (Tris-HCl) pH 8; 50 mM dithiothreitol (DTT); 2% sodium dodecyl sulfate (SDS)) for 5 min at 99 °C followed by sonication on the water at room temperature for 2 min. The magnetic beads, remaining in the lysate after cell isolation, were removed by centrifugation at 10,000× *g* RCF for 10 min and transferring supernatant to the new tubes. This step was repeated three times. Finally, the total protein concentration was measured using the Tryptophan Fluorescence (WF)-Based Assay [89].

### 4.3. MED FASP

Sample aliquots containing 70 µg of total protein were processed using the Multi-Enzyme Digestion Filter Aided Sample Preparation (MED FASP) method [90] with some recent modifications [91]. Briefly, proteins were consecutively digested overnight with endoproteinase LysC (Wako Chemicals, Neuss, Germany) and then with trypsin (Promega, Madison, WI, USA) 3 h. The enzyme to protein ratio was 1:50. Aliquots containing 10 µg of total peptides were desalted on C18-StageTips [92] and concentrated to a volume of ~5µL and were stored frozen at −20 °C until MS analysis.

### 4.4. LC-MS/MS 

Analysis of peptide mixtures was performed using a QExactive HF Mass Spectrometer (Thermo-Fisher Scientific). Aliquots containing a 5 µg of total peptides were chromatographed on a 50 cm column with 75 µm inner diameter packed C18 material (Dr. Maisch GmbH, Ammerbuch, Germany). Peptide separation was carried out at 300 nL/min for 90 min using two-step acetonitrile (ACN) gradient of 5–60% over the first 75 min and 60–95% for the following 15 min. The temperature of the column oven was 60 °C. The mass spectrometer operated in data-dependent mode with survey scans acquired at a resolution of 50,000 at m/z 400 (transient time 256 ms). Up to the top 15, most abundant isotope patterns with charge ≥ +2 from the survey scan (300–1650 m/z) were selected with an isolation window of 1.6 m/z and fragmented by HCD with normalized collision energies of 25. The maximum ion injection times for the survey scan and the MS/MS scans were 20 and 60 ms, respectively. The ion target value for MS1 and MS2 scan modes was set to 3 × 106 and 105, respectively. The dynamic exclusion was 25 s and 10 ppm. The mass spectrometry data have been deposited to the ProteomeXchange Consortium via the PRIDE partner repository [93] with the dataset identifier:PXD011683.

### 4.5. MS/MS Data Analysis

The spectra were searched using MaxQuant software (Max Planck Institute, Martinsried, Germany). A maximum of two missed cleavages was allowed. Carbamidomethylation of cysteines was set as a fixed modification. The minimum peptide length was specified to be seven amino acids. The initial maximal mass tolerance in MS mode was set to 7 ppm, whereas fragment mass tolerance was set to 20 ppm for HCD data. The maximum false peptide and protein discovery rate was specified as 0.01. Specific protein concentrations were calculated by the TPA [94] using raw intensity MQ output. Statistical analysis of proteomic data was conducted by Perseus software v.1.6.10.45 (Max Planck Institute for Biochemistry) [95]. FL samples were first grouped as follows: G1 vs. G3a, low-grade lymphoma (G1 and G2) vs. high-grade lymphoma (G3a), and high Ki-67 (range, 80% to 95%) vs. low Ki-67 (range, 25% to 60%). Next, samples were filtered for 100% of valid values in each group and Student’s *t*-tests were performed. As there were no significant differences in the protein abundances between the above-mentioned groups, all cases were assigned to the FL and further analyzed in comparison with the control group of normal B-cells. After further filtering for proteins detected in at least 75% of each group, 6724 proteins remained for further analysis. The absolute protein concentration was log2-transformed, and the separate values for each sample were compared with a two-sample T-test. Benjamini-Hochberg FDR procedure was used to calculate *p*-values, and *p* ≤ 0.05 was considered statistically significant. The cutoff values of 2-fold for up-regulated and 0.5- fold for down-regulated proteins (with *p*-value ≤ 0.05) between FL and normal samples were established. The fold change was calculated by dividing the mean TPA values of the lymphoma group with the mean TPA values of the control group.

### 4.6. Bioinformatic Analysis

The differentially abundant proteins with a *p*-value ≤ 0.05 were analyzed by open-source bioinformatics software platform Cytoscape version 3.8.0 [96] Functional interaction network analysis was performed using ClueGO (version 2.5.7), STRING, and cytoHubba Cytoscape plugins [97,98]. The DAPs were analyzed and visualized by ClueGO against KEGG, Reactome, and Wiki pathways (all updated on 8 May, 2020). For the enrichment of pathways, the two-sided hypergeometric test was used, showing only pathways with *p*-value ≤ 0.05. Bonferroni was selected as a statistical method. The Kappa-statistic score threshold was set to 0.4. GO tree intervals were set min level 2—max level 8, the number of associated proteins for the cluster was 3. The Venn diagrams were created with the web tool created by the Bioinformatics and Evolutionary Genomics group [99]. 

### 4.7. Flow Cytometry

Cells of interest were obtained by FNAB from 35 FL patients. Cells were incubated with monoclonal antibodies, Appendix A. Antigen expression was quantified by FACSCalibur and FACSCanto II cytometers (Becton Dickinson, Franklin Lakes, NJ, USA). Appendix A shows the results of CD9, CD79B, and ICAM1 validation with clinical characteristics of FL patients. Expression was categorized according to the percentages of positive cells into four groups, as followed: (−)—no expression (<20% of neoplastic cells); (+/−)—an antigen with weaker expression in FL cells compared to normal B- or T-lymphocytes in >20% to <100% of cells; (+)—highly expressed antigen in FL cells compared to normal B- or T-lymphocytes in 100% of cells; (+^)—very highly expressed antigen in FL cells compared to normal B- or T-lymphocytes in 100% of cells. All cases expressed pan-B antigens: CD19, CD20, and CD22, with median fluorescence intensity (MFI) of CD20 always higher than that of CD19. According to the results, the most common FCM immunophenotype of FL was as follows: CD45(+)^weaker^/CD20(+)^higher^/CD19(+)^weaker^/CD22(+)/CD10(+)/CD81(+)^higher^/BCL2(+)^higher^/BCL6(+)/CD44(+/−)^weaker/dim^/CD38(+)/λ(+)/IgD(−),IgM(+),IgG(−)/FMC7(+)^weaker^/CD5(−)/CD11c(−)/ CD23(−)/CD25(−)/CD43(−)/CD62L(+)/HLADR(+)^weaker/dim^/CD52(+)/CD49d(+)/CD200(−)/ CD305(−), and CD9(+/−)/ CD79B(+)/CD54(+)^higher^.

## 5. Conclusions

We presented the first report on the use of large scale proteomics-based analyses to describe the FL associated proteome at the level of thousands of proteins. A total of 1186 differentially abundant proteins, including 579 overrepresented proteins and 607 downrepresented proteins, were identified within the 29 clinical samples. We used the ClueGO App for Cytoscape to generate protein pathways and to create the network of pathways based on the KEGG, Reactome, and WikiPathways databases. Bioinformatic analyses identified several biological pathways as being particularly perturbed compared to normal B-cells. Overrepresented proteins were mainly related to the B-cell receptor signaling pathway and signaling by interleukins, which is in line with previously published studies. In turn, increased abundances were enriched in ECM-receptor interactions, regulation of complement cascade, and cell-cell communication.

To the best of our knowledge, the present study was the first aiming to determine the candidate proteins and pathways associated with FL using proteomics and bioinformatics analysis. Our dataset complemented previous findings substantially and revealed underappreciated numbers of proteins within larger networks that could lay the groundwork for therapeutic strategies as well as future studies on lymphoma adhesion and ECM-interactions. In the current era of personalized medicine and targeted therapies, the dysregulated signaling pathways and their hub proteins are quickly becoming mainstream in translational cancer research. Collectively, our work demonstrates a global view of follicular lymphoma related changes and provides novel insight into the molecular mechanisms of this malignancy, however the current data need to be validated in an independent dataset. 

## Figures and Tables

**Figure 1 cancers-13-00630-f001:**
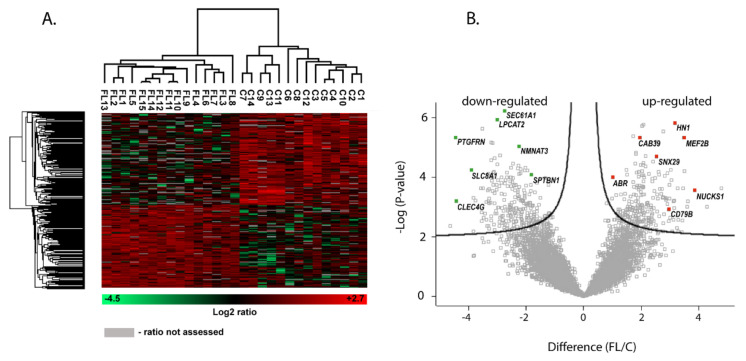
(**A**) Heat map representing the log2 (abundances) of the 454 differentially abundant proteins (DAPs) in 15 follicular lymphoma (FL) and 14 control samples (C). Generated by Perseus software after data normalization. The heat map was created based on the data clustering using Euclidean distance. Each row represents a differentially represented protein abundance and each column, a clinical sample. The color scale represents z-scores (log2 intensity). Significantly up-regulated proteins in FL are labeled in red, while significantly down-regulated proteins are labeled in green. Grey boxes indicate that the ratio was not assessed for a particular protein. (**B**) Volcano plot displays the main dysregulated proteins identified with a *p*-value ≤ 0.05 and ≥ 2-fold cutoff. Proteins were graphed by fold change (difference) and significance (-log *p*-value) using a false discovery rate (FDR) of 0.05 using the Perseus software (Max Planck Institute for Biochemistry, Martinsried, Germany).

**Figure 2 cancers-13-00630-f002:**
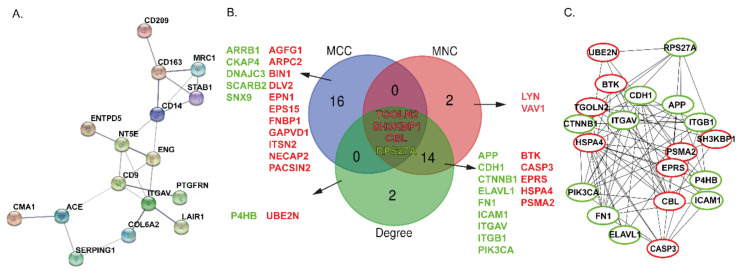
(**A**) Search Tool for the Retrieval of Interacting Genes/Proteins (STRING) analysis on the top down-regulated proteins with lymphoma/normal ratio ≤ 0.01. Edges represent protein-protein associations and the the thickness of the line indicates the strength of data support. Only connected nodes are shown. (**B**) Venn diagram illustrating the overlap of identified hub proteins between maximal clique centrality (MCC), maximum neighborhood component (MNC), and Degree analysis methods of cytoHubba. (**C**) Protein-protein interaction networks (PPINs) of the top 20 hub proteins selected from 1186 dysregulated proteins according to Degree algorithm from cytoHubba. Down-regulated proteins with fold change ≤ 0.5 are marked in green, while up-regulated proteins (fold change ≥ 2.0) are plotted in red, (*p* ≤ 0.05).

**Figure 3 cancers-13-00630-f003:**
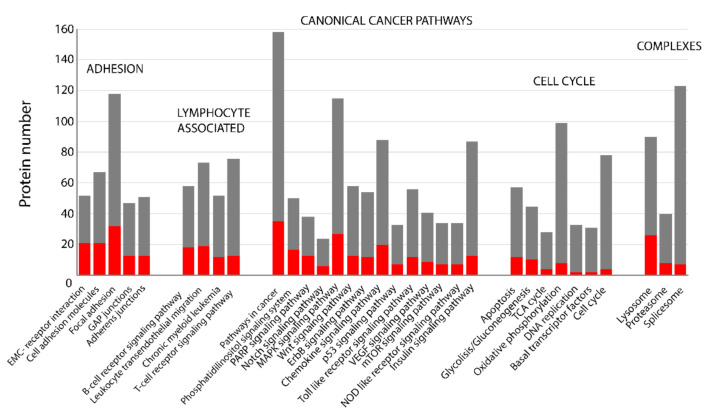
Kyoto Encyclopedia of Genes and Genomes (KEGG) pathway enrichment analysis for differentially abundant proteins (DAPs) in the quantified proteomes of FL. Down-regulated proteins with fold change ≤ 0.5 and up-regulated proteins with fold change ≥ 2.0 were used for the enrichment analyses of the KEGG pathways. The red and grey bars indicate the number of DAPs and all the identified proteins classified into different signaling pathways, and cellular processes, respectively. Performed in Perseus software, (*p* ≤ 0.05).

**Figure 4 cancers-13-00630-f004:**
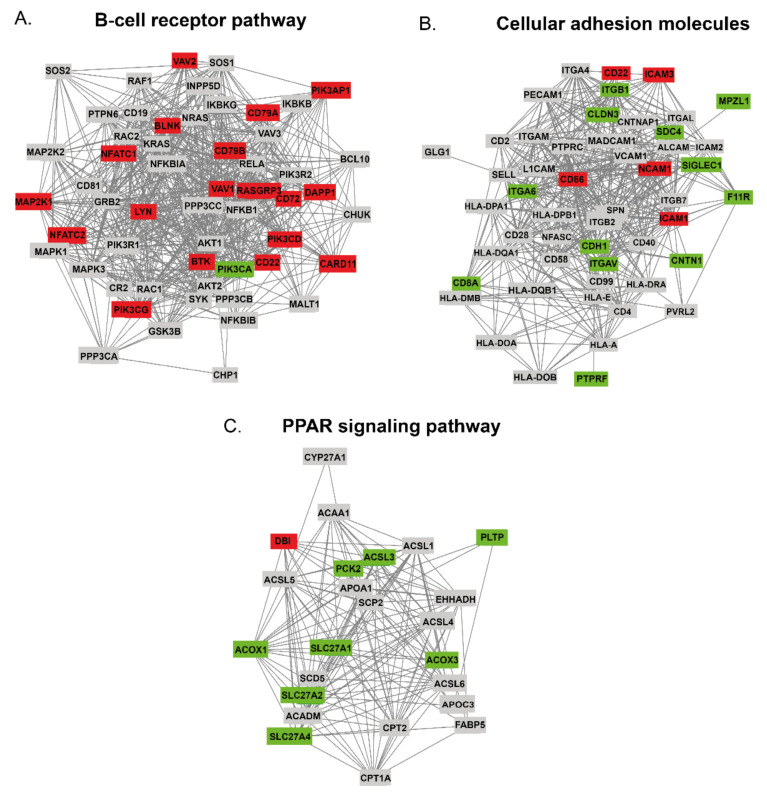
Functional networks of proteins involved in the following pathways: (**A**) B-cell receptor (BCR) signaling pathway (hsa04662), (**B**) cellular adhesion molecules (CAMs) pathway (hsa04514), (**C**) PPAR signaling pathway (hsa03320). Proteins identified by liquid chromatography-tandem mass spectrometry (LC-MS/MS) were qualified to certain pathways according to KEGG. Colored tags indicated the abundance increase (red) and decrease (green) in FL compared to normal B-cells. Integration of the PPIN was generated with Cytoscape. PPIN enrichment *p*-value was < 1.0 × 10^−16^ for all the networks.

**Figure 5 cancers-13-00630-f005:**
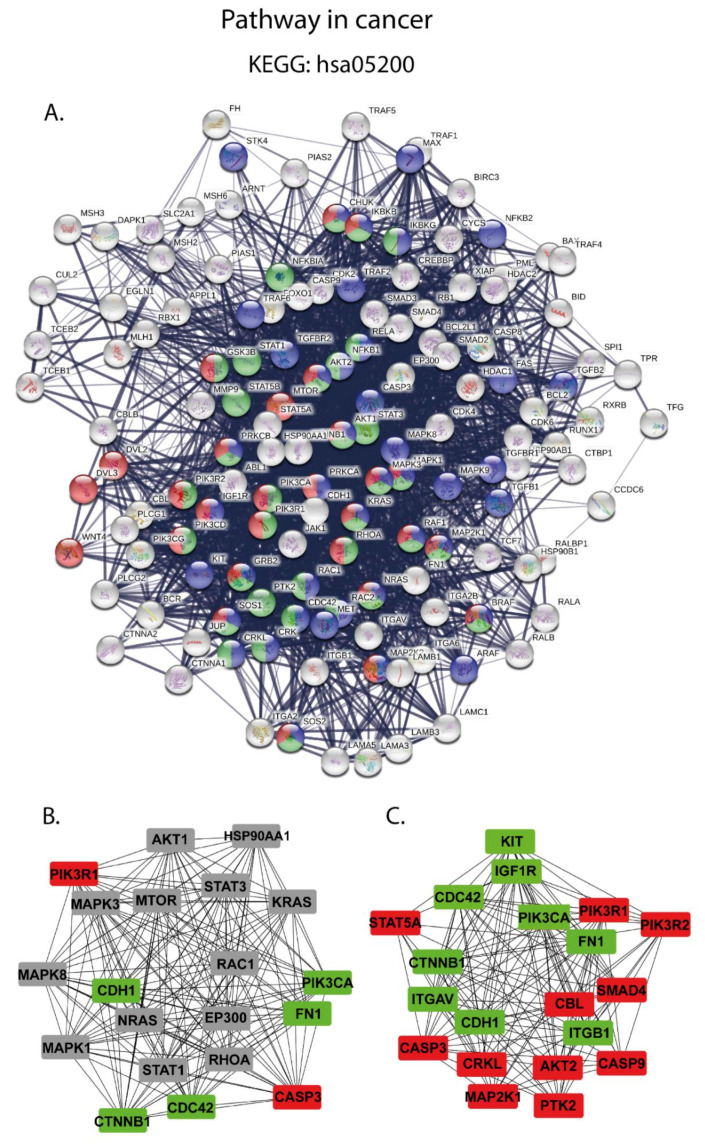
(**A**) Selected network of 138 proteins that were found in a functional screen for proteins involved in the pathway in cancer (KEGG: hsa05200). The network was created with the use of the STRING database. Edges connecting proteins represent interactions between two proteins, and the line thickness indicates the strength of data support. Red, blue, and green nodes represent the mTOR signaling pathway, MAPK signaling pathway, and chemokine signaling pathway, respectively. The overlapping areas indicate the shared proteins of any two or three groups. (**B**) Top 20 proteins with the highest degree of connectivity belonging to the pathway in cancer in the entire experiment. (**C**) Top 20 DAPs with the highest degree of connectivity. Colored tags indicated the abundance increase (red) and decrease (green) in FL compared to normal B-cells.

**Figure 6 cancers-13-00630-f006:**
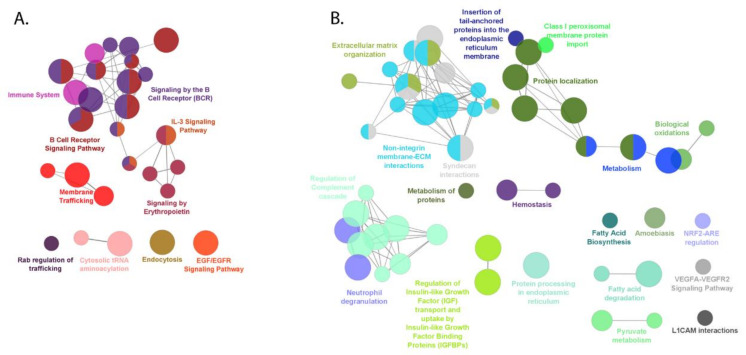
Network view for (**A**) up-regulated and (**B**) down-regulated protein pathways in FL. Functional interaction network analysis was performed using ClueGO Cytoscape App. The DAPs were mapped to REACTOME, KEGG, and WikiPathways (all updated 18.05.2020). Nodes (circles) indicate the pathway function groups, while edges represent connections between the nodes. The node size represents the term enrichment significance. The length of each edge is linked with relevancy between two processes. Each node color indicates each class that they belong to, *p* ≤ 0.05. The overlapping areas indicate the shared pathways of any two or three groups. The up- and down-regulated proteins, which contributed to the enriched pathways are listed in Appendix A, respectively.

**Table 1 cancers-13-00630-t001:** The top 20 (**A**) up- and (**B**) downrepresented proteins in follicular lymphoma (FL). Appendix A includes details of all 1168 differentially abundant proteins (DAPs), FL/control ratios, and Student’s *t*-test statistics.

	**(A)**			
**Gene Name**	**Protein Name**	**Accession**	**Fold Change**	**−Log *p*-Value**
*NUCKS1*	Nuclear ubiquitous casein and cyclin-dependent kinase substrate 1	Q9H1E3	36.535	4.47
*SLC14A1*	Urea transporter 1	Q13336	34.949	3.37
*GPALPP1*	GPALPP motifs-containing protein 1	Q8IXQ4	24.970	1.69
*ADPRH*	[Protein ADP-ribosylarginine] hydrolase	P54922	21.681	3.80
*CD72*	B-cell differentiation antigen CD72	P21854	21.000	2.29
*PRDM15*	PR domain zinc finger protein 15	P57071	20.148	1.71
*HMGN1*	Non-histone chromosomal protein HMG-14	P05114	19.527	3.59
*MEF2B*	Myocyte-specific enhancer factor 2B	Q02080-2	19.051	5.72
*PLEKHG2*	Pleckstrin homology domain-containing family G member 2	Q9H7P9	16.283	2.01
*DBNL*	Drebrin-like protein	Q9UJU6-2	14.342	1.61
*NAGPA*	N-acetylglucosamine-1-phosphodiester alpha-N-acetylglucosaminidase	Q9UK23	14.081	1.84
*SPTY2D1*	Protein SPT2 homolog	Q68D10	13.958	1.81
*SEMA4A*	Semaphorin-4A	Q9H3S1	13.311	3.33
*CCDC43*	Coiled-coil domain-containing protein 43	Q96MW1	12.680	3.75
*ARPP19*	cAMP-regulated phosphoprotein 19	P56211	12.429	1.83
*RASGRP3*	Ras guanyl-releasing protein 3	Q8IV61	11.771	3.17
*DDT*	D-dopachrome decarboxylase;D-dopachrome decarboxylase-like protein	P30046	11.570	2.49
*SP100*	Nuclear autoantigen Sp-100	P23497	11.558	3.38
*GGACT*	Gamma-glutamylaminecyclotransferase	Q9BVM4	11.222	3.47
*RRM2B*	Ribonucleoside-diphosphate reductase subunit M2 B	Q7LG56	11.201	3.67
	**(B)**			
**Gene Name**	**Protein Name**	**Accession**	**Fold Change**	**−Log *p*-Value**
*ITGAV*	Integrin alpha-V	P06756	0.045	3.46
*NT5E*	5-nucleotidase	P21589	0.047	2.93
*PLTP*	Phospholipid transfer protein	P55058	0.048	1.81
*CNTN1*	Contactin-1	Q12860	0.053	2.14
*CD163*	Soluble CD163	Q86VB7	0.056	6.11
*CLDN3*	Claudin-3	O15551	0.059	4.36
*CD14*	Monocyte differentiation antigen CD14	P08571	0.063	4.02
*CD9*	CD9 antigen	P21926	0.066	3.96
*CRAT*	Carnitine O-acetyltransferase	P43155	0.066	1.77
*STAB1*	Stabilin-1	Q9NY15	0.067	4.27
*MYO1B*	Unconventional myosin-Ib	O437	0.068	4.28
*TMEM119*	Transmembrane protein 119	Q4V9L6	0.070	1.84
*PTGFRN*	Prostaglandin F2 receptor negative regulator	Q9P2B2	0.070	5.26
*SERPING1*	Plasma protease C1 inhibitor	P05155	0.071	4.23
*MACROD1*	O-acetyl-ADP-ribose deacetylase MACROD1	Q9BQ69	0.075	5.54
*RAB34*	Ras-related protein Rab-34	Q9BZG1	0.077	2.58
*ACE*	Angiotensin-converting enzyme	P12821	0.078	2.72
*ENTPD5*	Ectonucleoside triphosphate diphosphohydrolase 5	O75356	0.080	2.77
*ENG*	Endoglin	P17813	0.081	3.03
*IGHG3*	Ig gamma-3 chain C region	A0A087WXL8	0.081	3.76

## Data Availability

Proteomic data is uploaded to the ProteomeXchange Consortium via the PRIDE partner repository (http://proteomecentral.proteomexchange.org/cgi/GetDataset) with the dataset identifier: PXD011683.

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
