# Peer review of "Large-Scale Proteomic Analysis of Follicular Lymphoma Reveals Extensive Remodeling of Cell Adhesion Pathway and Identifies Hub Proteins Related to the Lymphomagenesis"

_cancers, 2021, doi:10.3390/cancers13040630_

Round 1

Reviewer 1 Report

The paper addresses a timely and relevant clinical issue. The work is technically sound and was performed using state-of-the-art proteomics approaches. Conclusions are justified in the presented data. However, the authors should consider the modification and improvement of the way data are presented. In general, the paper is overloaded with graphs (party with low-resolution pictures), which include information to some extent covered in the text. Please consider including some of them as supplementary material. Presentation of fewer pictures with higher quality would make the paper concise and help to highlight the most important findings.

Specific comments.

1) One cannot say “unsupervised hierarchical clustering of the top 454 differentially abundant proteins was performed in Perseus software and revealed distinct protein profiles between lymphoma and control samples (page 4)”, because once differentially proteins were selected this is no more “unsupervised” analysis (hence, nice separation of groups is warranted) – please rephrase. I would be interesting two see results of actual “unsupervised” clustering based on all identified proteins.

2) The quality of graphics in Figure 2 should be improved: protein names are barely readable in Panel A and C. Moreover, the sequence of panels B and C seem confused concerning the text description (see lines 223-226).

3) Please make the title/legend to Figure 3 more clear (the information “what it is” seems more important than “how it was obtained” in this place). 

4) The quality of graphics in Figure 4 should be improved: protein names are barely readable. What is the rationale to show panels D-F? Graphics and text description seem redundant. 

5) Protein names are barely readable in Panel D and E in Figure 5 (which are partly redundant with text).

6) Please provide a stronger rationale for the selection of 3 proteins for validation (they represent <1% of DAPs). Text description (lines 441-456 page 13) and graphics (Fig. 7B) are redundant.

7) A part of the discussion is placed in the Results section (e.g., references 26-36). On the other hand, details of results are not necessarily recalled in the Discussion section. Please consider rephrasing this section to make it concise and focused, which would help to emphasize the most original findings.

8) Supplementary Table S4 with clinical characteristics should be checked: tumor diagnosis seems missed.

Reviewer 2 Report

The authors present a large-scale proteomic analysis of follicular lymphoma (FL) samples (n=15) and identified 1186 differentially abundant proteins between FL and control samples, causing an extensive remodeling of several molecular pathways, including the B-cell receptor signaling pathway, cellular adhesion molecules, and PPAR pathway. They also identify Hub proteins playing a role in follicular lymphomagenesis.  

Overall the manuscript is well written and the methodology seems appropriate. I have a few comments for the authors’ consideration.

Comments:

  1. Page 2, line 78: 25-30% rate of histological transformation in FL is fairly high. Please correct this. Although the survival is lower than indolent NHLs, it is not 1-2 years. This needs to be corrected as well.  
  2. It is stated that the cases of FL were diagnosed via FNA, however, the grading is difficult when there is limited tissue. How accurate was the grading (grade 1, 2, or 3A)? How many cases of FL were 1 vs 2 vs 3A. What was the Ki-67% of those with grade 3A?
  3. The authors note that the PIK3CA was down-regulated, while other phosphoinositide 3-kinases (PI3Ks) were frequently overexpressed. How would you explain this? Please elaborate.
  4. While I agree that the current work by authors is impressive and provides additional insights into the follicular lymphomagenesis, I would recommend toning down the conclusion as the current data needs to be validated in an independent dataset.

Round 2

Reviewer 1 Report

No further concerns.

Author Response

Dear Reviewer, thank you very much for for your time and accepting our changes in the orignical submission. 

With best wishes

Kamila DuÅ›-Szachniewicz with Co-authors. 

Wroclaw Medical University

Poland

Reviewer 2 Report

The majority of my comments have been addressed, but I have few additional comments jotted below.

1. You have not addressed my previous comment on the decreased survival of HT patients? It is not 1-2 years as you outline in the introduction. There are several factors that go into this including the time from FL to HT (<1 year versus >1 year), presence of molecular aberrations, etc.

2. Thanks for providing the Ki-67%. Yes, this reviewer is fully aware that the Ki-67% can be lower than that of grades 1-2, but it is imperative to know this information to tease out those with high Ki-67% as those behave differently biologically.

I note there are several patients with high Ki-67% (atleast 6 patients with Ki-67% >=80%). These patients need to be precluded from the analysis as they do not behave like your typical indolent NHLs. In fact, some even treat these patients differently. The results you present cannot be accurately interpreted with these patients included in the analysis. 

Author Response

Please see the attachment. Additional Excell file for Reviewer with results of Student's t-test was included in Supplementary Material. 

Round 3

Reviewer 2 Report

The queries have been appropriately addressed.